# An Injury-like Signature of the Extracellular Glioma Metabolome

**DOI:** 10.3390/cancers16152705

**Published:** 2024-07-30

**Authors:** Yooree Ha, Karishma Rajani, Cecile Riviere-Cazaux, Masum Rahman, Ian E. Olson, Ali Gharibi Loron, Mark A. Schroeder, Moses Rodriguez, Arthur E. Warrington, Terry C. Burns

**Affiliations:** 1Department of Neurological Surgery, Mayo Clinic, Rochester, MN 55905, USA; ha.yooree@mayo.edu (Y.H.); krajani@alumni.wfu.edu (K.R.); riviere-cazaux.cecile@mayo.edu (C.R.-C.); masum.rahman@bmc.org (M.R.); ian.olson1@northwestern.edu (I.E.O.); gharibiloron.ali@mayo.edu (A.G.L.); schroeds@charter.net (M.A.S.); warrington.arthur@mayo.edu (A.E.W.); 2Department of Neurological Surgery, Northwestern University, Chicago, IL 60208, USA; 3Department of Neurology, Mayo Clinic, Rochester, MN 55905, USA; rodriguez.moses@mayo.edu

**Keywords:** microdialysis, CNS injury, glioma

## Abstract

**Simple Summary:**

In this study, we performed microdialysis in patient-derived xenografts of glioma, intending to evaluate the extracellular metabolic impacts of temozolomide, a component of standard-of-care treatment. Unexpectedly, the most striking finding in our analyses was a tumor-like metabolic signature induced by catheter insertion into a normal brain. The development of a glioma-like extracellular metabolome in the days following microdialysis catheter placement into a non-tumor-bearing brain suggests particular care is needed to accurately discriminate the longitudinal extracellular metabolomic impacts of experimental therapies from those of evolving injury-associated changes.

**Abstract:**

Aberrant metabolism is a hallmark of malignancies including gliomas. Intracranial microdialysis enables the longitudinal collection of extracellular metabolites within CNS tissues including gliomas and can be leveraged to evaluate changes in the CNS microenvironment over a period of days. However, delayed metabolic impacts of CNS injury from catheter placement could represent an important covariate for interpreting the pharmacodynamic impacts of candidate therapies. Intracranial microdialysis was performed in patient-derived glioma xenografts of glioma before and 72 h after systemic treatment with either temozolomide (TMZ) or a vehicle. Microdialysate from GBM164, an IDH-mutant glioma patient-derived xenograft, revealed a distinct metabolic signature relative to the brain that recapitulated the metabolic features observed in human glioma microdialysate. Unexpectedly, catheter insertion into the brains of non-tumor-bearing animals triggered metabolic changes that were significantly enriched for the extracellular metabolome of glioma itself. TMZ administration attenuated this resemblance. The human glioma microdialysate was significantly enriched for both the PDX versus brain signature in mice and the induced metabolome of catheter placement within the murine control brain. These data illustrate the feasibility of microdialysis to identify and monitor the extracellular metabolome of diseased versus relatively normal brains while highlighting the similarity between the extracellular metabolome of human gliomas and that of CNS injury.

## 1. Introduction

Gliomas are primary brain tumors, the most common of which are aggressive infiltrative astrocytomas [1]. Treatment of these aggressive tumors with standard-of-care maximally safe surgical resection, temozolomide (TMZ), and radiation only extends life expectancy to a median of 15 months [2]. Metabolic plasticity provides an avenue for gliomas to adapt to microenvironmental and therapeutic stress, leading to increased interest in metabolically targeted therapies [3,4]. To date, relatively few studies have attempted to understand the in situ longitudinal metabolic impacts of candidate therapies through the analysis of metabolic changes within live tumor tissue [5]. Microdialysis enables the sampling of extracellular fluid compositions through a perfused catheter with a semipermeable membrane. Small soluble analytes, such as metabolites, able to cross the semipermeable membrane (“dialyzable” analytes) can be sampled longitudinally from the extracellular compartment of diseased and healthy tissues [6]. We and others have previously reported a reproducible extracellular metabolome across a diverse array of gliomas, including elevated D-2-hydroxyglutarate (D-2-HG) within gliomas harboring a mutation in isocitrate dehydrogenase 1 (IDH1) [7], the most common glioma subtype in most young adult patients [8].

Patient-derived xenografts (PDXs) in athymic nude mice retain phenotypic and molecular characteristics of patient tumors [9]. While the method for intratumoral microdialysis in murine models is well established [10], no studies to our knowledge have tested if the global extracellular metabolome of human glioma is replicated within orthotopic (intracranial) PDXs, nor have these models been leveraged to discern the metabolic impacts of therapy. To that end, we asked whether microdialysis could be utilized to discriminate the extracellular metabolome of a glioma PDX and the metabolic impacts of TMZ. Using an IDH-mutant glioma PDX (GBM164), we identified a robust metabolic signature of the glioma extracellular microenvironment characterized by many of the changes previously reported in human gliomas compared with adjacent brains [7,11]. Importantly, this signature closely resembled changes induced over time following catheter placement into a normal brain. Finally, analysis of the extracellular glioma metabolome of human patients demonstrated more robust enrichment for the injury-associated metabolome than that of the glioma PDX itself. 

## 2. Materials and Methods

### 2.1. Ethics Statement

All animal experiments were reviewed and approved by the Institutional Animal Care and Use Committee (IACUC), Mayo Clinic, Rochester, MN.

### 2.2. Intracranial Implantation 

In this study, we utilized GBM164, generated from an IDH-mutant grade 4 astrocytoma and maintained as a patient-derived xenograft (PDX) [12]. Of note, this cell line was established and published prior to the publication of the 2021 CNS WHO guidelines. As such, although CNS WHO 2021 astrocytomas (IDH-mutant; grade 4) are no longer regarded as glioblastomas, for consistency with the established literature, the originally published nomenclature of “GBM164” is utilized in this manuscript [9]. Orthotopic xenografts were generated as previously reported [13]. Briefly, on the day of cranial implantation, cells were dissociated using trypsin (TrypIE, Catalog # 12563011, ThermoScientific, Waltham, MA, USA) prior to suspension in PBS at a concentration of 100,000 cells/µL. Female athymic nude mice (Charles River Laboratories, Wilmington, MA, USA) were utilized for all experiments. The mice were anesthetized using a mixture of ketamine/xylazine (100 mg/kg ketamine and 10 mg/kg xylazine; IP). After placing the animals in the stereotactic frame with the line between the bregma and lambda horizontal, the heads were prepared with betadine and alcohol prior to a midline incision. A burr hole was drilled 1 mm anterior and 2 mm lateral to the bregma to a depth of 2 mm. Tumor cells were then intracranially injected at a rate of 1 µL/min for 3 min using a 26G non-coring Hamilton syringe. The syringe was left in place for 3 min to minimize the risk of cellular reflux. Control animals intended for subsequent analysis of the “Brain” were treated identically but implanted with the same volume of saline rather than tumor cells. Bone wax was applied to the burr hole, and the wound was sutured with 4-0 Vicryl. A triple antibiotic was applied to the incision line. All mice received water supplemented with children’s ibuprofen for 48 h prior to and after surgery for analgesia. The mice were monitored daily for a week, and sutures were removed at 10 days post-operatively. 

### 2.3. Microdialysis Implantation and Collection

Forty-five days after tumor implantation (approximately 60% of the time to moribund for GBM164), magnetic resonance imaging (MRI) was performed with a Bruker Avance 300 mHz, 7 Tesla, vertical-bore nuclear magnetic resonance spectrometer (Bruker Biospin, Billerica, MA, USA) to confirm the tumor size and location. The protocol for intracranial PDX glioma microdialysis has been previously described in [10]. Briefly, the animals were anesthetized, prepped, and positioned as above before intracranial implantation. A guide cannula for the microdialysis probe (model CXG-02, Amuza, San Diego, CA, USA) was implanted into the prior injection site. Dummy probes (model CXD(T)-2, Amuza, San Diego, CA, USA) were placed into the guide cannula. Healing was permitted for 24 h prior to obtaining the baseline microdialysate sample. Microdialysis was performed by replacing the dummy probe with a pre-primed brain microdialysis probe (model CX-I-2-02, Amuza, San Diego, CA, USA; 50 kDa; 2 mm membrane). Microdialysis was performed at 1 µL/min using a 2.5 mL Hamilton gastight syringe (Hamilton Reno, NV, USA) placed in a series ESP101 pump (Amuza, San Diego, CA, USA). To facilitate the collection of equilibrated samples without dilution from the catheter dead space, microdialysate collection began at 30 min and was continued for 2 h, after which the dummy probe was replaced. The animals then received TMZ (50 mg/kg; PO) or a vehicle immediately after baseline microdialysate collection. Microdialysis was then repeated 72 h later (see Figure 1 for the experimental outline). The microdialysis samples were stored at −80 °C.

### 2.4. Untargeted Metabolomic Analysis

Untargeted metabolomic analysis was performed via ultra-performance liquid chromatography–tandem mass spectrometry (UPLC-MS/MS; Metabolon, Inc., Morrisville, NC, USA). This combination of liquid chromatography and mass spectrometry allows for the separation of different components within a mixture and the identification of each separated component. Two fractions were evaluated with reverse-phase (RP) UPLC-MS/MS with positive ion-mode electrospray ionization (ESI), another fraction with RP/UPLC-MS/MS with negative ion-mode ESI, and a fourth with hydrophobic interaction chromatography (HILIC) UPLC-MS/MS with negative ion-mode ESI. Throughout processing, multiple controls were analyzed with the experimental samples, including a pooled matrix sample, extracted water samples, and QC standards for instrument performance monitoring and chromatographic alignment. The relative standard deviation (RSD) was measured for instrument variability control, and the overall process variability was determined by the median relative standard deviation for all endogenous metabolites. The experimental samples were randomized and processed with QC samples spaced evenly among the injections. Metabolites were identified by comparing the data with Metabolon’s library which includes information on the retention index (RI), mass-to-charge ratio (*m*/*z*), and chromatographic data of molecules. Peaks were quantified based on the area under the curve. In total, 226 metabolites were identified. Of these, 152 metabolites were present in at least 90% of samples (*n* = 33) and included in subsequent analyses.

### 2.5. Ranked Metabolite Lists

To determine which metabolites defined the extracellular environments of either the tumor or the brain, the mean fold change between tissue types was calculated for each metabolite. The means of the normalized peak areas detected at baseline in GBM164 (*n* = 9) were divided by the means of those at baseline in the brain (*n* = 12) and then ranked from highest to lowest (fold change: tumor/brain). Metabolites with a higher rank (tumor/brain) were more closely associated with the glioma, while metabolites with a lower rank were associated with the brain.

To determine the time-dependent metabolic impacts of the vehicle or TMZ administration on each animal, the fold changes of the normalized peak area were calculated for each metabolite within 72 h post-vehicle or post-TMZ treatment compared with the animal’s pre-treatment baseline and then ranked from highest to lowest. Metabolites that were only detected at 72 h were manually assigned a rank based on the magnitude of the raw peak area. For example, consider a mouse in which maltose and tartronate were undetected at baseline but present at 72 h with peak areas of 585,283 and 90,433, respectively. A true fold change could not be calculated due to the absence of the metabolite at baseline, as the values could not be divided by zero. As such, in this case, maltose was assigned rank “1” and tartronate assigned “2” based on their magnitudes at 72 h. The ranks of each metabolite for each animal were then averaged (*n* = 3) to determine an overall ranked list of metabolite changes for each comparison.

Mean-ranked lists were also created to allow comparisons between the different treatment groups. For example, the vehicle-treated tumor versus brain signature was created by dividing the mean normalized peak area of each metabolite across all tumors at 72 h post-vehicle administration by the mean across all brains 72 h post-vehicle administration. The same process was used to create the TMZ-treated tumor versus brain signature. These ranked lists were then used to identify conserved patterns via enrichment analysis. 

### 2.6. Enrichment Analysis

Gene set enrichment analysis (GSEA) is a rank-based method of quantifying bioinformatic enrichment that is usually employed in the context of gene or protein-based analyses. However, this method can also be repurposed for metabolomic studies [7]. Using GSEA, a custom library of “feature sets” can be queried against a ranked list of analytes to determine where the analytes from each feature set fall along this ranked list. If the analytes in a set are most frequently found at the top of the input ranked list, this is known as “positive enrichment”; if they fall at the end of the ranked list, this is known as “negative enrichment”. The GSEA software calculates a normalized enrichment score (NES), *p*-value, and false discovery rate (FDR) for each analyte set in the library. For each inquiry, a custom metabolite library was created using the top and bottom 35 metabolites for each ranked metabolite list of interest. We used enrichment analysis (GSEA version 4.3.2) in these studies to determine the relative positive or negative enrichment of each mouse for other mice that were subjected to similar or different experimental conditions. To achieve this, the full ranked metabolite list for each mouse (.rnk file) was run against a metabolite library created from the experimental cohort (.gmx file). We also used EA to determine how the tumor versus brain signature changed in response to variables like 72 h of catheter injury with or without the administration of TMZ and to assess the enrichment for the human glioma extracellular metabolome. 

### 2.7. Statistics

Analyses were performed using the normalized peak areas (median = 1 across metabolites for each sample). The normal distribution of metabolites across samples was tested via the D’Agostino–Pearson test using GraphPad PRISM 10. For paired analyses, Wilcoxon signed-rank tests for non-parametric distributions were performed on metabolites using MetaboAnalyst. Non-parametric Mann–Whitney U tests were applied to unpaired analyses. Volcano plot cut-offs were set at fold change (FC) ≥ 1.5 and *p* ≤ 0.05. Graphs were generated using Graphpad PRISM 10. Enrichment analysis was performed using GSEA 4.3.2 (gene set enrichment analysis), repurposed for metabolomic analysis. FDR ≤ 0.05 was considered statistically significant for enrichment analysis.

## 3. Results

### 3.1. Metabolic Signature of GBM164, an Orthotopic Model of High-Grade Glioma

We first asked how microdialysis could be utilized to evaluate the extracellular metabolome of a representative glioma PDX model. Mice were stereotactically intracranially injected with either GBM164 cells, an established glioma patient-derived xenograft harboring an IDH mutation, or saline (the control). After confirming the presence of a tumor via MRI, the mice underwent microdialysis catheter insertion approximately 7 weeks after tumor or control implantation. Microdialysate collection was performed in the tumor or brain 24 h after catheter placement. Microdialysis enables the direct sampling of analytes within the extracellular space and can be deployed in both tumors and normal brain tissue [10]. Untargeted metabolomic analysis was performed on the Metabolon platform [14]. 

To characterize the global metabolome of the glioma PDX, we compared the microdialysate collected at baseline from orthotopic models of GBM164 versus the control mice. We evaluated the average fold change in metabolites between the tumor (*n* = 9) and brain (*n* = 12) and ranked the metabolites from highest to lowest fold changes (tumor vs. brain), which, from here on, comprised the metabolic signature of GBM164 versus brain. The relative abundances of the 100 most differentially abundant metabolites between the tumor and brain are depicted for each animal in Figure 2A. Using fold-change thresholds (tumor/brain) of >1.5 and *p* < 0.05, 38 metabolites were found to be significantly more abundant in tumors, while 33 metabolites were more abundant in brain microdialysate (Figure 2B).

The most significant tumor-associated metabolites included 2-hydroxyglutarate (FC = 20.5×; *p* = 6.8 × 10^−6^), a well-characterized oncometabolite of IDH-mutant tumors [11,15], N,N,N-trimethyl-5-aminovalerate (FC = 2.9×; *p* = 6.8 × 10^−6^) and asparagine (FC = 2.6×; *p* = 6.8 × 10^−6^) (Figure 2C). The most significant brain-associated metabolites included alpha-ketoglutaramate (FC = 0.4×; *p* = 1.3 × 10^−5^), methylmalonate (FC = 0.4×; *p* = 2.7 × 10^−5^), and N-acetylglutamine (FC = 0.4×; *p* = 4.8 × 10^−5^).

### 3.2. Metabolic Impacts of Catheter Insertion over Time

Unlike tissue sampling, which can typically be performed only at a single time point, microdialysis enables the longitudinal sampling of the tumor microenvironment. Since the insertion of a microdialysis catheter is expected to induce some modest localized tissue injury that could evolve over time, we, next, asked how the extracellular metabolome may vary between the “baseline” sample obtained 24 h after catheter placement and a second sample obtained 72 h thereafter. A ranked list (1 to 152) ordered by fold change was generated for each animal by comparing the difference in the abundance of metabolites between baseline and 72 h (72 h/baseline). Ranking the animal-specific fold-change lists based on the average rank across the three animals revealed a consistent pattern of differentially abundant metabolites between baseline and 72 h across the animals (Figure 3A). The top and bottom 10 metabolites based on the average rank are shown adjacent to the heatmap, including numerous amino acids that were more abundant after catheter insertion. The similarity across animals suggested a time-dependent impact of catheter insertion on the extracellular metabolome of non-tumor-bearing brains. These findings were further confirmed by enrichment analysis between the pairings of each animal’s ranked lists, demonstrating the significant enrichment of the 72 h vs. baseline metabolome across the animals (FDR < 0.05 for all comparisons; Figure 3A inset). To determine if any metabolites were significantly altered at 72 h versus baseline within each mouse, we performed Wilcoxon signed-rank tests of the metabolite fold changes between baseline and 72 h. The paired analysis of the cohort (*n* = 3) yielded no significant fold changes in the metabolites at cut-offs of FC > 1.5× and *p* < 0.05, suggesting that while the overall catheter-induced metabolic signature was significantly conserved across mice, three animals per group yielded insufficient power to demonstrate significant impacts of catheter placement at the level of individual metabolites. Since injury causes inflammation, which is also prominent in tumors, we asked whether the injury signature induced over time could be related to the metabolic signature of glioma. We performed enrichment analysis between the post-catheter insertion brain and the baseline tumor versus brain signatures and observed significant enrichment (NES = 2.27; FDR < 1 × 10^−3^; Appendix A). Moreover, baseline brain (versus tumor) metabolites were negatively enriched in the 72 h versus baseline signature (NES = −2.13; FDR < 1 × 10^−3^; Appendix A).

Having established that the extracellular metabolome of a non-tumor-bearing brain in the days following catheter insertion was enriched for tumor-like changes, we next asked how the metabolic signature of the tumor (GBM164) itself is impacted over the same period after microdialysis catheter placement. We applied the same analyses to vehicle treatment in tumors and observed a slightly less consistent extracellular metabolic impact of catheter insertion at 72 h versus baseline on tumors across the cohort (Figure 3B), with significant enrichment between animals (the NESs are depicted in bold if FDR < 0.05) in some but not all comparisons (Figure 3B inset). Enrichment of the post-catheter insertion tumor was not significant for either the baseline tumor or brain signature (Appendix A).

### 3.3. Metabolic Impacts of Catheter Insertion over Time in the Presence of TMZ

Having observed the time-dependent metabolic impacts of catheter insertion on the tumor and brain, we, next, asked how the administration of TMZ may impact the observed extracellular metabolomes of the brain and tumor over time. We, again, compared the brain microdialysate at 72 h with baseline in animals treated with TMZ immediately after the completion of the baseline microdialysis. Once again, a similar metabolic pattern was observed between animals after TMZ treatment based on the order of ranked metabolites between the time points (Figure 3C). However, the enrichment for the 72 h vs. baseline metabolic signature was only significant between two of three TMZ-treated animals (FDR < 0.05; Figure 3C inset). Paired analysis using Wilcoxon signed-rank tests, again, identified no significantly differentially abundant metabolites between the brain at 72 h after TMZ administration compared with baseline. We then evaluated the tumor extracellular microenvironment at 72 h after TMZ treatment compared with baseline, noting an absence of significantly differentially abundant metabolites between the time points as well as some heterogeneity between animals (Figure 3D).

### 3.4. Impact of TMZ or Catheter Insertion on the Glioma versus Brain Metabolome

We, next, asked if the overall metabolic signature of the glioma versus brain at baseline remained similar after catheter-induced injury, with or without TMZ administration. Ranked metabolite lists were generated based on the average fold change of the normalized metabolite abundance between the “tumor” versus “brain” at the 72 h time point (*n* = 3/group). This was repeated for cohorts of animals treated with either the vehicle or TMZ (Figure 4A). Enrichment analysis was then performed to determine how these post-injury glioma versus brain signatures, with or without TMZ, each compared with the baseline glioma signature. Significant positive enrichment was observed across each comparison, suggesting that neither the delayed impact of catheter insertion (“vehicle”) nor 72 h of TMZ exposure fundamentally abrogated the metabolic signature of the glioma versus brain at baseline (Figure 4B–D).

### 3.5. Comparison with Human Glioma Microdialysate

Finally, we asked how the murine microdialysis findings compared with those of human glioma. We recently reported the metabolome of intra-operatively sampled glioma microdialysate [7]. Eight patients in that cohort had paired catheters located in both the enhancing tumor and the brain adjacent to the tumor. By creating enhancing tumor versus brain ranked lists for each patient, we performed enrichment analysis to evaluate the similarity of the human glioma signatures to the (a) PDX versus brain, (b) catheter-induced injury (72 h vehicle vs. baseline in the brain), and (c) TMZ-treated injury (72 h TMZ vs. baseline in the brain) ranked metabolite lists. Significant enrichment (FDR < 0.05) was noted in eight out of eight patients for both the PDX versus brain and catheter-induced injury signatures. In contrast, significant enrichment was only noted in two out of eight patients for the TMZ-treated injury signature (Figure 5A), demonstrating that TMZ alters the metabolic impact of catheter injury in a way that lessens its resemblance to a tumor. 

Unexpectedly, eight out of eight patients’ tumor signatures were significantly more enriched (*p* = 1.0 × 10^−4^) for the murine injury signature than the PDX glioma signature itself. The patients’ tumor signatures were also more significantly enriched for both the injury and PDX glioma signatures than they were for the injury signature with TMZ (*p* < 1.1 × 10^−4^; *p* = 1.1 × 10^−3^, respectively; Figure 5A).

We performed similar enrichment analyses for the PDX glioma, catheter-induced injury, and TMZ-treated injury metabolic signatures using an average patient tumor versus brain ranked metabolite list generated from the average fold change across the eight patients. Significant enrichment was observed when comparing the average patient tumor versus brain ranked list with the top metabolite sets of the average injury signature (NES = 3.62; FDR < 1 × 10^−3^) and PDX signature (NES = 2.87; FDR < 1 × 10^−3^) (Figure 5B). We also performed an enrichment analysis between the average patient tumor versus brain ranked list and the top metabolite sets of the average signature of TMZ-treated injury (72 h TMZ vs. baseline in the brain), but this analysis did not yield a significant enrichment score.

The metabolites at the leading edge of enrichment between the injury signature and the average patient tumor ranked list are listed in a table with the corresponding fold changes of the average PDX vs. brain signature, the 72 h vehicle vs. baseline injury signature, the 72 h TMZ vs. baseline injury signature, and the average patient tumor versus brain signature (Figure 5C). Of note, despite these analyses being guided by the results in mouse microdialysate, the fold changes between the tumor and brain in the human samples far exceeded those observed in the mouse tumors.

Solid tumors often have fenestrated vasculature [16] and are known to have a “wound-like” signature. Our prior findings in human high-grade gliomas demonstrated that blood–brain barrier disruption significantly contributed to the extracellular metabolomic signature [7]. As such, to determine whether the catheter-induced injury signature could be, in part, due to plasma-derived metabolites, we performed enrichment analysis on the catheter-induced injury signature using a bloody vs. clean human CSF ranked list derived from pooled samples in the human study [7]. Significant enrichment was observed (NES = 2.08; FDR < 0.01; Appendix A). We, then, also looked for enrichment between the signature of murine tumors and that of bloody vs. clean human CSF. The significant enrichment observed (NES = 1.82; FDR < 0.05; Appendix A) suggested that plasma-derived metabolites also contribute to the tumor signature. Of note, the enrichment for these plasma-derived metabolites was not as strong as the enrichment for the injury signature with the tumor itself, suggesting that these plasma-derived extracellular metabolites may contribute to some, but not all, of the catheter-induced injury and murine tumor metabolic signatures.

## 4. Discussion

We performed microdialysis using an orthotopic model of glioma to compare the extracellular metabolic impacts of the tumor versus brain and the induced extracellular alterations after catheter placement during vehicle or TMZ treatment. Our findings revealed that (1) GBM164 has a strong metabolic signature, including an elevated abundance of 2-HG in the tumor versus brain, consistent with the findings in the microdialysate of human IDH-mutant gliomas [7,11]; (2) the metabolic evolution of a catheter-induced brain injury signature at 72 h is reminiscent of both human and mouse tumor versus brain signatures; (3) the tumor extracellular metabolome of glioma is highly resilient, with neither catheter-induced injury nor TMZ administration inducing considerable changes to the overall tumor versus brain signature; and (4) TMZ abrogates the metabolomic resemblance of the catheter injury to the human tumor versus brain signature.

Microdialysis has been used to understand the pharmacokinetic and pharmacodynamic impacts of systemically administered agents as well as agents delivered locally via reverse microdialysis [17,18]. We originally intended to use microdialysis to evaluate the extracellular metabolic impacts of temozolomide, a component of standard-of-care treatment for patients with glioma. However, the most striking finding in our analyses was the tumor-like metabolic signature induced by catheter insertion into a non-tumor-bearing brain. CNS injury causes the upregulation of vascular endothelial growth factor (VEGF), which promotes angiogenesis and astroglial proliferation in the damaged region [19,20]. Similarly, VEGF is constitutively activated in the tumor neovasculature, inducing leakage from plasma and contributing to a wound-like signature [16,21]. The catheter-induced injury signature was significantly enriched for that of the tumor, which we have previously shown to be enriched for plasma-derived metabolites [7]. Future studies may query if the normalization of VEGF-A-induced vascular disruption could impact the metabolic signature of injury and tumors. That said, although, bevacizumab is an anti-VEGF-A antibody currently approved for recurrent glioblastoma that improves symptoms associated with increased BBB disruption [22], elective neurosurgical procedures are infrequently pursued after Avastin, given its negative impacts on wound healing [23].

The observation that catheter insertion alone induced a reproducible metabolic response over time may be relevant to the interpretation of studies evaluating the pharmacodynamic impacts of therapy via the extracellular metabolome [24]. Many microdialysis studies are performed in the early post-operative setting [25,26,27], during which a dynamic post-operative extracellular metabolome may require diligent consideration of internal controls to ensure changes are associated with the therapy rather than recent surgery. If, indeed, the glioma extracellular metabolome mimics a CNS injury, as our data would suggest, catheter insertion may have a less obvious impact, given the already elevated levels of injury-associated metabolites in the glioma-associated microenvironment. Moreover, inevitable heterogeneity within the tumor microenvironment [28,29] may require sampling from multiple locations to increase confidence in the reproducibility of the observed metabolic impacts.

The mechanism via which TMZ alters the extracellular metabolic signature of injury warrants further evaluation. To our knowledge, no study has specifically investigated the effect of TMZ on vascular permeabilization or wound-healing processes. However, the impacts of CNS injury include microglial and astrocyte proliferation [30], and neuroinflammation can induce blood–brain barrier disruption [31,32]. Future studies could evaluate the impact of TMZ on the production of inflammatory cytokines associated with blood–brain barrier disruption, the expression of VEGF, and measures related to vascular integrity, such as tight-junction proteins (i.e., claudins and occludins). Although speculative, an alternative explanation could be changes in the expression of L-type amino acid transporter 1 (LAT1), a neutral amino acid transporter. LAT1 expression can be upregulated in the context of spinal cord injury [33] and is correlated with poor survival in patients with GBM [34].

The limitations of this study include the lack of an immediate post-insertion baseline since the cannula and healing dummy were placed 24 h prior to microdialysis to allow time for vascular re-annealing. An additional confounding factor may include the removal and replacement of the microdialysis catheter at each time point with the intervening usage of a healing dummy, which could also potentially have induced minor disruptions in the microenvironment. The minimal sample size of this study (*n* = 3) likely underpowered our ability to ask questions about the metabolic impact of TMZ within the murine tumors, given that gliomas are characterized by significant intratumoral heterogeneity [28,29]. Enrichment analysis can, nonetheless, be used to discern similarities in biology, even when groups are underpowered to demonstrate statistically significant changes in individual metabolite levels between time points. Moreover, only one PDX line was utilized; as such, it is possible that some findings may be cell-line specific, including the relatively modest impacts of TMZ administration on the overall murine tumor signature. Nevertheless, comparisons with eight genomically diverse human gliomas were performed to evaluate the applicability and reproducibility of the other three main findings of this study with more power. Future studies in other PDX cell lines will be needed to further inform our finding that neither injury nor TMZ administration induced changes to the murine tumor signature. Further work will also be needed to discern the kinetics and longevity of tumor-like metabolic changes beyond the 72 h time point. Similar studies may also incorporate radiotherapy in their experimental designs, as chemoradiation is the current standard-of-care treatment for newly diagnosed glioblastoma [2]. To date, extracellular pharmacodynamic impacts have only been evaluated in single-agent experiments and the context of recurrent disease [18,35]. The ability to discern the pharmacodynamic impacts of effective therapies may provide a tool with which to improve individualized treatment algorithms.

## 5. Conclusions

Ongoing clinical studies utilizing intra-operative high-molecular-weight microdialysis have revealed a conserved extracellular metabolic signature across genetically diverse high-grade gliomas [7]. This is consistent with the idea that therapeutically targetable convergent metabolic vulnerabilities may be present in human glioma, a hypothesis actively being pursued via microdialysis in clinical trials [24]. Our finding of the longitudinal metabolic impact of catheter insertion suggests that special consideration may be required to deconvolute the impacts of catheter-induced injury from that of drug-induced metabolic pharmacodynamic changes.

## Figures and Tables

**Figure 1 cancers-16-02705-f001:**
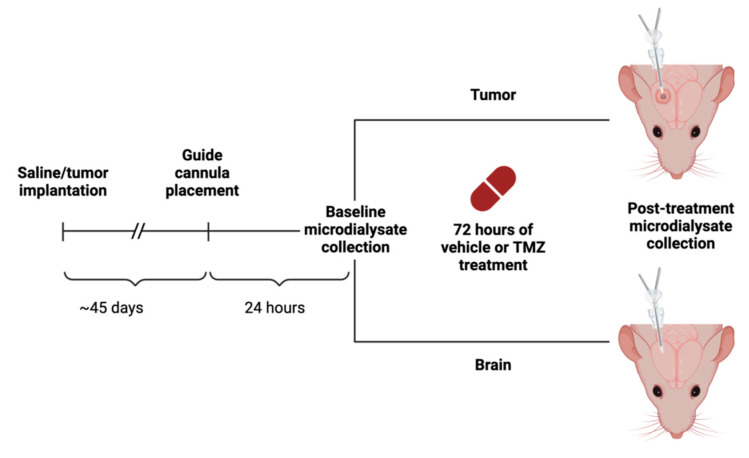
Schematic of experimental design. Following TMZ or vehicle (1X PBS) administration, microdialysate was collected at two different time points, once at baseline and after 72 h. With two variables of tissue type (tumor and brain) and treatment option (TMZ or vehicle), there were four experimental conditions. This figure was created with BioRender.com.

**Figure 2 cancers-16-02705-f002:**
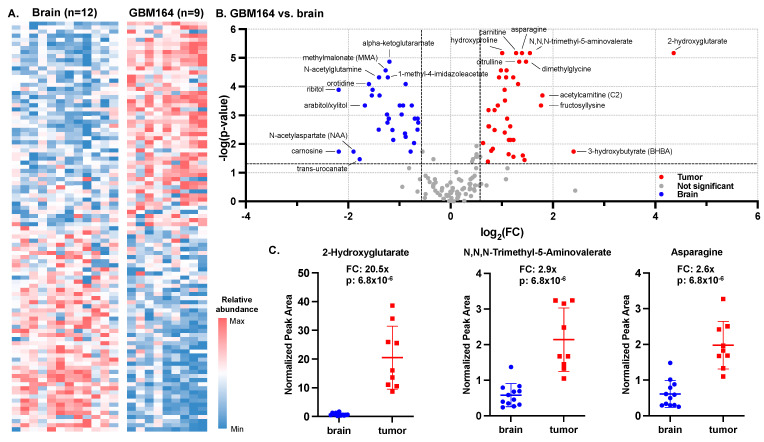
Microdialysis of IDH-mutant high-grade glioma versus normal brain. (**A**) Heat map of top and bottom 50 differentially abundant raw metabolite peak values for both brain and tumor based on fold changes between the means. (**B**) Volcano plot of fold changes and Mann–Whitney U test of fold changes calculated between IDH-mutant PDX and brain at baseline (*n* = 9 for tumor; *n* = 12 for brain) were used to construct a tumor versus brain volcano plot (cut-offs for significance marked as dashed lines: *p*-value ≤ 0.05; FC ≥ 2). (**C**) Fold changes and *p*-values (based on Mann–Whitney U test) of 3 most significantly altered metabolites.

**Figure 3 cancers-16-02705-f003:**
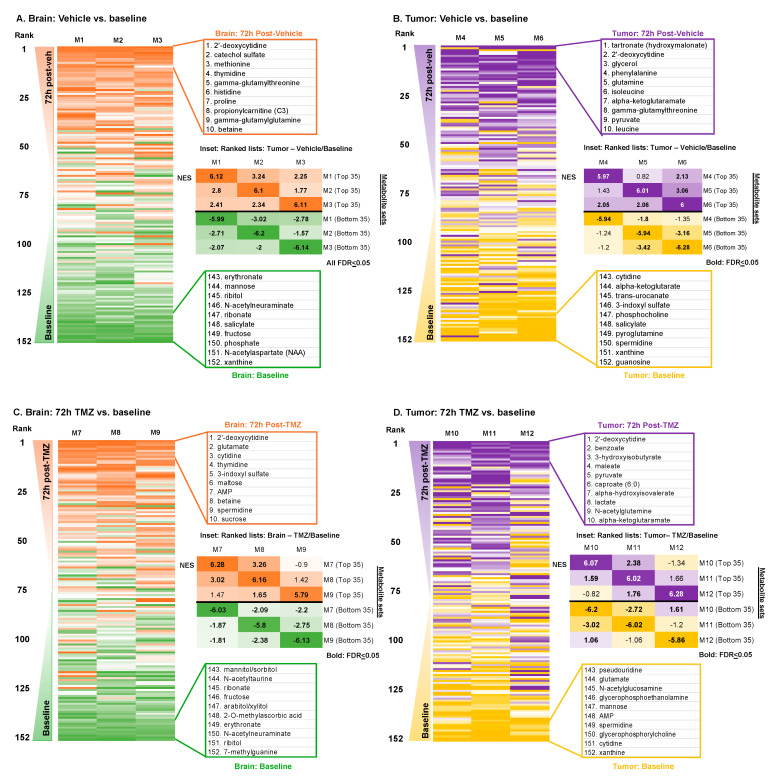
Metabolic signatures of vehicle and TMZ in brain and tumor (72 h vs. baseline) (**A**) For each mouse with a vehicle-treated brain, the fold changes of the peak area were calculated for each metabolite (72 h vs. baseline) and ranked from highest to lowest. These ranked lists were then ordered based on the average of the experimental cohort (*n* = 3). Inset: NES values from enrichment analyses performed between the total ranked metabolite list of each individual mouse and a library composed of the top and bottom 35 metabolite lists of each mouse. (**B**) The same analyses performed in A were applied to mice with vehicle-treated tumors. (**C**) For each mouse with a TMZ-treated brain, the fold changes of the peak area were calculated for each metabolite (72 h vs. baseline) and ranked from highest to lowest. These ranked lists were then ordered based on the average of the experimental cohort (*n* = 3). Inset: NES values from enrichment analyses performed between the total ranked metabolite list of each individual mouse and a library composed of the top and bottom 35 metabolite lists of each mouse. (**D**) The same analyses performed in C were applied to mice with TMZ-treated tumors.

**Figure 4 cancers-16-02705-f004:**
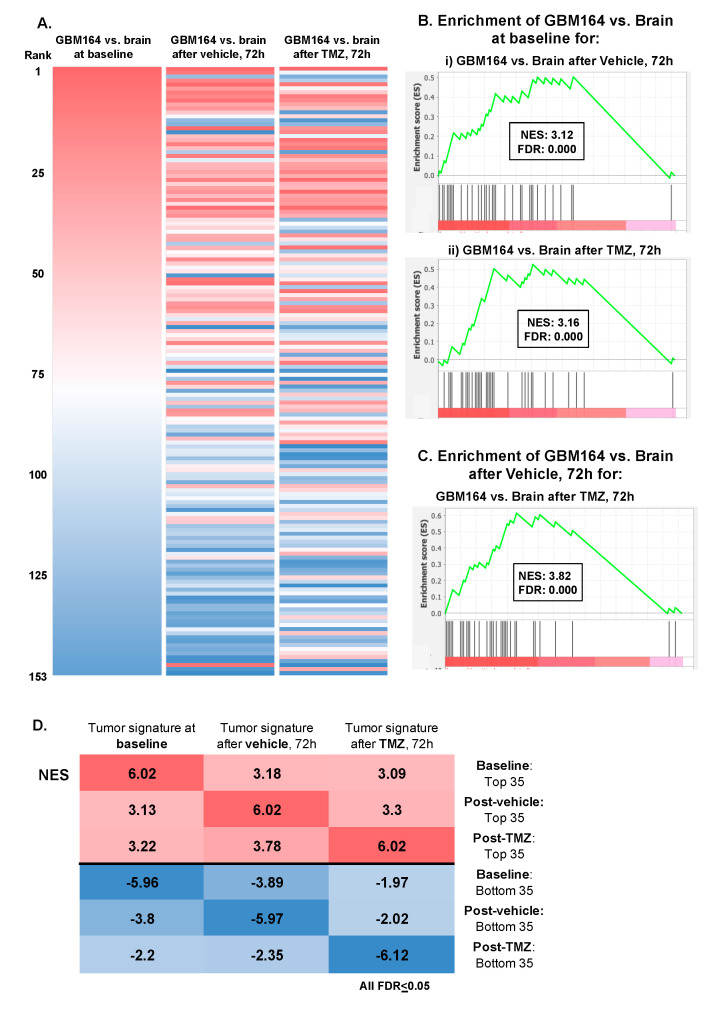
Defining the extracellular metabolome of tumor (**A**) Ranked lists of the GBM164 vs. brain signature at baseline, 72 h after vehicle administration, and 72 h after TMZ treatment were created by evaluating the average fold changes in metabolites between brain and tumor at the respective time points and then ranking metabolites from highest to lowest fold changes. These ranked lists were ordered based on the GBM164 vs. brain signature at baseline. (**B**) Plots of enrichment analysis between the baseline GBM164 vs. brain ranked list and metabolite sets of (**i**) GBM164 vs. brain after 72 h of vehicle administration and (**ii**) GBM164 vs. brain after 72 h of TMZ treatment. (**C**) Plot of enrichment analysis between the ranked list of GBM164 vs. brain after 72 h of vehicle administration and the top 35 metabolites of GBM164 vs. brain after 72 h of TMZ treatment. (**D**) NES values from enrichment analyses performed between each ranked list and a library composed of the top and bottom 35 metabolites of every list.

**Figure 5 cancers-16-02705-f005:**
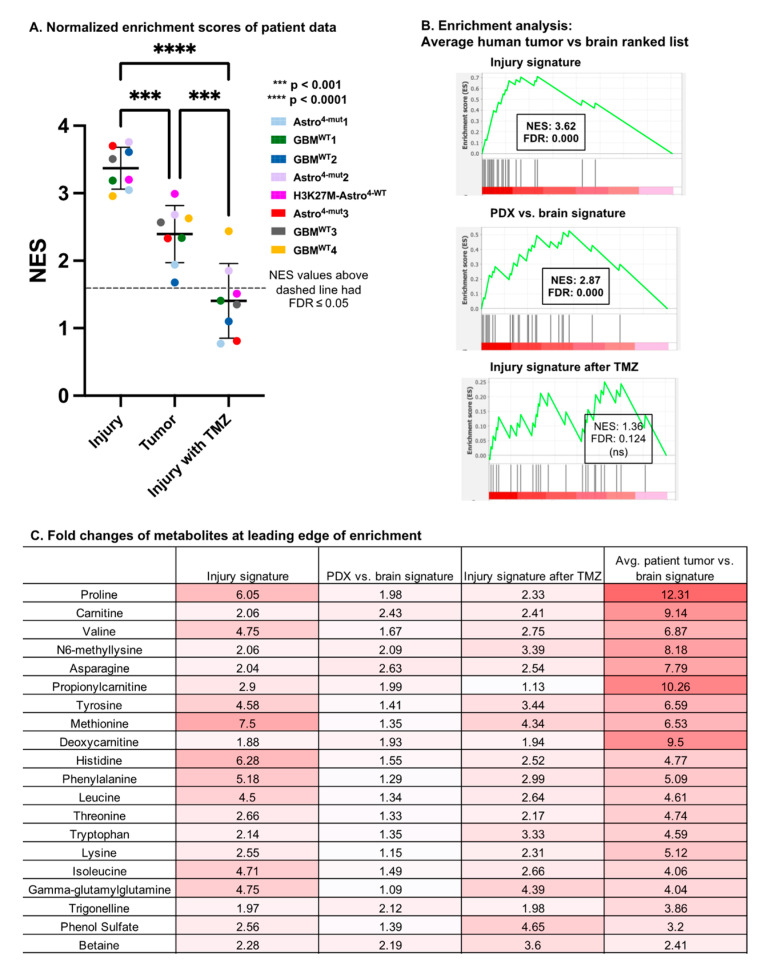
Enrichment with intra-operatively acquired human glioma microdialysate (**A**) Enrichment analysis was utilized to determine the enrichment of eight human patients’ enhancing tumor vs. brain ranked lists for the metabolite sets of catheter-induced injury, PDX vs. brain, and catheter-induced injury in the setting of TMZ. Positive normalized enrichment scores (NESs) indicate metabolic similarities to these signatures. ANOVA and pairwise comparisons of normalized enrichment scores calculated between the average patient glioma signature and the ranked lists of catheter-induced injury (72 h vs. baseline in vehicle-treated brain), the glioma PDX signature at baseline, and the injury signature after TMZ treatment (72 h vs. baseline in TMZ-treated brain) are shown. All NES values above the dashed line had FDR ≤ 0.05. (**B**) Enrichment plots of average patients’ enhancing tumor vs. brain ranked list and the top 35 metabolites of the injury signature, glioma PDX signature at baseline, and the injury signature after TMZ. (**C**) Fold changes of metabolites at leading edge of enrichment in injury signature and average patient tumor vs. brain signature.

## Data Availability

The raw data are contained within the Appendix A.

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
