# Peer review of "An Injury-like Signature of the Extracellular Glioma Metabolome"

_cancers, 2024, doi:10.3390/cancers16152705_

Round 1

Reviewer 1 Report

Comments and Suggestions for Authors

The authors attempt to evaluate extracellular metabolic impact of temozolomide in patient derived intracranial orthotopic xenograft glioma model. Not only they found the mice engrafted with GBM164 display greater level of 2-HG compared to their control counterpart, but they also discover the metabolic signature of tumor brain resemble catheter induced brain injury, which can be abrogated by temozolomide treatment. While the findings are attractive, there is only one cell line (GBM164) used in the orthotopic model, hence it is unable to eliminate cell lines induced bias. It is recommended to repeat and include additional one glioma cell line in the in-vivo mouse study.   

Reviewer 2 Report

Comments and Suggestions for Authors

In this study, the authors performed intratumoral vs. intrabrain tissue microdialysis metabolite sampling using an orthotopic PDX model of IDH-mutated grade 4 astrocytoma (i.e., GBM164) in order to capture and compare the extracellular metabolic signatures of tumoral vs. normal brain tissues as well as the induced metabolic alterations after catheter placement during vehicle or temozolomide (TMZ) treatment. The study is fairly interesting and also important for its clinical relevance for intra-operative/post-operative metabolite sampling during the surgical and/or chemoradiation treatment for high-grade glioma patients. The key findings of the study were: (i) the metabolic signature of the GBM164 model was found to be consistent with the readings from microdialysate samples of human IDH-mutant gliomas, which further qualifies the relevance of this PDX model, (ii) the catheter-induced brain injury signature was found to be reminiscent of both human and mouse tumor signatures when compared to the normal brain tissue signature, (iii) the tumor extracellular metabolic signature reading did not appear to change in the presence of catheter-induced injury  nor TMZ administration when compared to the normal brain tissue signature,  and (iv) the TMZ treatment appeared to acutely (i.e., over 72 hours) abrogate the metabolomic resemblance of the catheter injury to the human/mouse tumor when compared to the normal brain tissue signature. While these findings are quite informative and have translational potential, I am not entire sure about their interpretation by the authors. Or at least some of these interpretations need to come with additional explanations.  Accordingly, I have a number of comments for the authors as follows. 

1.      In my opinion, the injury metabolic signature prompted by catheter placement is most likely due to the extravasation of plasma-derived metabolites. The similarity between this injury signature and the tumor extracellular metabolic signature (both for the PDX model and the clinical data analysis) is probably the result of the leakiness of the blood-brain-barrier (BBB) in the tumor beds. As found by the authors, the correlation between the catheter injury and the tumor signatures are even stronger when doing these analyses using metabolites sampled from patients with enhancing tumors (i.e., tumors with highly disrupted BBBs). The leakiness of the tumor microvasculature in solid tumors is a long-established phenomenon known to generate wound healing signatures, and this notion has been revisited and reinforced many times since the 1986 landmark essay by Harold Dvorak in NEJM. For this reason, the finding by the authors that connects the catheter-related injury with the tumor metabolic signature is rather expected and not necessarily novel in itself. However, I agree that this observation could have an important impact on how microdialysate samples are collected and further analyzed/interpreted in the clinic. For instance, it would be interesting to test whether the administration of bevacizumab (Avastin) dose could also abrogate this catheter-related injury signature as well as its confounding effect when interpreting metabolic data in the clinic. Bevacizumab has a pronounced stabilization effect on vascular permeability via VEGF trapping. Perhaps the authors could consider this approach for a future experiment. More importantly, the potential effect of the alteration of vascular permeability due to catheter insertion and the role of plasma-related metabolites in the interpretation of their data should be discussed by the authors in the Discussion section of their manuscript.

2.        Related to the above, it is unclear to me how a single dose of TMZ in the mouse tumor model (at the equivalent of 150 mg/m2 in humans) could alter (abrogate) the extracellular metabolic injury signature resulting from catheter insertion. What is the mechanism of this? Is this alkylation agent known to have an effect on vascular permeabilization whatsoever? Or otherwise on delaying wound healing? To my knowledge, this has not been studied. By extension, is TMZ known to have a stabilizing effect on the leakiness of the tumoral BBB? Do the authors have an explanation/speculation for this observed effect with TMZ that is worth adding to their discussion section? Also, while I believe this effect has merit and needs to be further investigated,  it is a little unclear to me why TMZ was tested in this study in the first place. First off, the sampling of microdialysate in a clinical scenario will most probably happen in the peri-operative setting (i.e., long before the chemoradiation protocol is initiated). Second, the treatment of high-grade gliomas is chemoradiation, meaning that there is very little, if any place for TMZ to be used as a monotherapy in these patients.   

3.     Lastly, I would recommend the authors to revisit the Results section of their manuscript and rewrite some of the parts for better clarity. For instance, the interpretation by the authors of some of the included data (and this is especially true for Figs. 3 and 4 in my opinion) is not very obvious.  It would certainly help if the authors would provide more explanations in the text regarding how they analyzed and interpreted these data for the benefit of the potential reader. 

Reviewer 3 Report

Comments and Suggestions for Authors

The manuscript: "An injury-like signature of the extracellular glioma metabolome" by Yooree Haet al presents a study on the metabolome of patient-derived xenografts of glioma in order to evaluate the extracellular metabolic impacts of the drug temozolomide and to compare with previous data obtained on human gliomas study. The authors used microdialysis technique which allows the sampling of the extracellular fluid within brain and brain tumors with the surrounding tissue. Next, by liquid chromatography coupled with mass spectrometry method the collected samples separated into different components and further, the components were identified and ranked. Finally, by applying the Gene Set Enrichment Analysis for collected microdialysates, the study determined how the murine tumor versus brain metabolome's signature changed within 72 hours of catheter injury with/without the administration of TMZ. The results from murine experiments were compared with the authors' previous published results obtained on human gliomas metabolome study.

The authors showed that there is a distinct metabolic signature of IDH-mutant glioma patient-derived xenograft versus normal brain, and that the murine tumor recapitulates the metabolic features observed in human gliomas. Moreover, the metabolic changes observed in the mice brain 72 hours after catheter-induced injury were similar to the metabolic signatures seen in both human and mouse brain tumors, but the TMZ administration attenuated the catheter-induced injury metabolic changes. This suggests that the mere act of inserting a catheter into the brain can induce metabolic alterations in time, which can resemble those observed in the tumor microenvironment, even in the absence of an actual tumor.

The manuscript is well written, all sections present detailed and scientifically proven information. The conclusions well emphasized the importance of this work.

The results presented in the study should be useful for clinical trials, and special care is needed for data interpretation to discern between the effects of catheter-induced injury and those determined by drug pharmacodynamics on metabolites in the tumor microenvironment.

Round 2

Reviewer 1 Report

Comments and Suggestions for Authors

The authors have address the all the suggestions. The manuscript is ready and can be accepted in the present form. I have no further comments.